https://doi.org/10.1038/s41467-022-31132-7　　**OPEN**

# CXCL4 synergizes with TLR8 for TBK1-IRF5 activation, epigenomic remodeling and inflammatory response in human monocytes

Chao Yang [1], Mahesh Bachu[1], Yong Du [1], Caroline Brauner[1], Ruoxi Yuan[1], Marie Dominique Ah Kioon [1], Giancarlo Chesi[1], Franck J. Barrat [1,2,3] & Lionel B. Ivashkiv [1,2,4 ✉]

Regulation of endosomal Toll-like receptor (TLR) responses by the chemokine CXCL4 is implicated in inflammatory and fibrotic diseases, with CXCL4 proposed to potentiate TLR responses by binding to nucleic acid TLR ligands and facilitating their endosomal delivery. Here we report that in human monocytes/macrophages, CXCL4 initiates signaling cascades and downstream epigenomic reprogramming that change the profile of the TLR8 response by selectively amplifying inflammatory gene transcription and interleukin (IL)−1β production, while partially attenuating the interferon response. Mechanistically, costimulation by CXCL4 and TLR8 synergistically activates TBK1 and IKKε, repurposes these kinases towards an inflammatory response via coupling with IRF5, and activates the NLRP3 inflammasome. CXCL4 signaling, in a cooperative and synergistic manner with TLR8, induces chromatin remodeling and activates de novo enhancers associated with inflammatory genes. Our findings thus identify new regulatory mechanisms of TLR responses relevant for cytokine storm, and suggest targeting the TBK1-IKKε-IRF5 axis may be beneficial in inflammatory diseases.

[1] HSS Research Institute and David Z. Rosensweig Genomics Research Center, Hospital for Special Surgery, New York, NY, USA. [2] Immunology and Microbial Pathogenesis Program, Weill Cornell Medicine, New York, NY, USA. [3] Department of Microbiology and Immunology, Weill Cornell Medicine, New York, NY, USA. [4] Department of Medicine, Weill Cornell Medicine, New York, NY, USA. ✉email: IvashkivL@hss.edu

A subset of Toll-like receptors comprised of TLR3/7/8/9 is expressed on endosomes and is important for sensing nucleic acids (NAs) from internalized pathogens; TLR3 senses double-stranded RNA, TLR7/8 sense single-stranded RNA, and TLR9 senses double-stranded DNA. Activation of endosomal TLRs (eTLRs) induces type I IFN and cytokine production and accordingly the eTLRs play a key role in host defense against various viruses including herpes simplex virus and SARS-CoV-2[1]. eTLRs can also sense extracellular NAs that are released by dying or netting cells and are internalized and delivered to endolysosomes for eTLR activation. Thus, eTLRs are important sensors of tissue damage and cell death that occur in sterile inflammation. Importantly, effective activation of eTLRs by self-NAs requires binding of extracellular self-NAs by molecules often termed chaperones that protect NAs from degradation and facilitate endocytosis and delivery to endolysosomal compartments[2–4]. Such potentiation of eTLR activation can be accomplished by anti-NA autoantibodies[5], or by cationic proteins (such as HMGB1) and short α-helical cationic antimicrobial peptides (β-defensins, LL37) which bind DNA or RNA via charge:charge interactions[2,3,6–9]. A role for potentiation of eTLR responses by NA-binding chaperones in disease pathogenesis is well-established, for example by anti-NA antibodies in SLE and NA-binding peptide LL37 in psoriasis[4,5,10,11].

eTLR signaling requires prior activation by proteolytic cleavage and translocation to an acidified endolysosomal compartment to bind NA ligands. TLR3 utilizes the signaling adapter TRIF, whereas TLR7/8/9 utilize MyD88; ligand binding initiates a proximal signaling cascade involving kinases IRAK1/4 and adapters TRAF3/6[4,12,13] with downstream activation of interferon regulatory factor (IRF) transcription factors which then induce *IFN* expression. TLR3 activates IRF3 in multiple cell types, whereas TLRs 7 and 9 activate primarily IRF7 via IRAK1 and IKKα in plasmacytoid DCs and TLR7/8/9 activate IRFs1/5 in conventional DCs and macrophages. IRF5 was recently shown to be recruited to endolysosomes by an adapter termed TASL[14,15] where IRF5 is phosphorylated by IKKβ or TBK1/IKKε (homologous kinases that act together, with typically a dominant role for TBK1)[16–18]. eTLRs also activate kinases TAK1 and IKKα/β, which are linked with downstream activation of NF-κB and inflammatory target genes. As IRF1 and IRF5 activate both IFN and inflammatory genes in cooperation with NF-κB[13,19], these IRFs can mediate integrated gene responses downstream of eTLRs. The balance between activation of IRF-IFN and NF-κB-inflammatory responses is modulated in part by localization and transit of ligand-receptor complexes along the endolysosomal pathway, which can be affected by the NA-binding chaperones described above[10,20].

The endosomal TLRs are differentially expressed and have distinct functions in different immune cell types in a species-specific manner[4,10,21–24]. TLR7 and TLR9 are co-expressed in pDCs and B cells and linked to IFN responses and B cell activation. In mouse cells, TLR9 is also expressed in myeloid cells, whereas TLR8 appears to be nonfunctional in immune cells as it lacks sequences that are necessary for ssRNA recognition[25,26]. In human, TLR8 is expressed in myeloid cells and can mediate IFN and inflammatory responses[27–29]. Although TLR8 has been relatively understudied as it is nonfunctional in mouse cells, an important role is emerging in human disease pathogenesis. Human TLR8 (hTLR8) and hTLR7 are highly expressed in rheumatoid arthritis (RA) synovial macrophages and contribute to production of pathogenic inflammatory cytokines[26,30–33], and hTLR8 is aberrantly expressed in pDCs in systemic sclerosis (SSc) where it drives pathogenic IFN production[34]. TLR8 gain of function somatic mutations result in an inflammatory environment associated with T cell activation and abnormal B cell

differentiation[35]. Transgenic mice expressing human TLR8 in myeloid cells develop a multiorgan inflammatory syndrome and increased collagen-induced arthritis[26]. Thus, it is important to better define mechanisms of hTLR8 activation and its coupling to inflammatory gene activation.

CXCL4 is a short cationic peptide that was originally described as a chemokine (termed platelet factor 4; PF4) that is a major product of platelets and is also produced by other immune cell types including pDCs and macrophages[36]. CXCL4 is present at high concentrations at sites of tissue injury and inflammation (up to 10 μg/ml in IBD or SSc[6,37,38]) and has been implicated in various inflammatory diseases including atherosclerosis, IBD, SSc, and RA[37,39,40]. CXCL4 augments TLR9-induced IFN production in pDCs, promotes differentiation of macrophages[41], and polarization of cDCs into a high cytokine-producing and pro-fibrotic phenotype[34,42–44]. The ability of CXCL4 to promote inflammation and/or fibrosis was difficult to understand based on its function as a chemokine. This was addressed by a paradigm-shifting study[6] showing that CXCL4 can function independently of its receptor CXCR3 by binding self-NAs as a chaperone to facilitate cellular uptake and increase eTLR activation. Similar to LL37 and other NA-binding chaperones, CXCL4 binds to cell surface proteoglycans (PGs)[36,45,46], which facilitates internalization of bound NAs. Different from LL37, CXCL4 alone can activate signaling in myeloid cells, which occurs independently of chemokine receptors[36]. Instead, CXCL4 signaling is thought to be mediated by low-affinity binding to cell surface proteoglycans which facilitates CXCL4 interaction with signaling receptors on the plasma membrane[47]. In contrast to its chaperone function, the role of CXCL4-induced signaling in cooperating with and augmenting eTLR responses is not known.

Here we characterize the pro-inflammatory impact of CXCL4, and mechanisms by which CXCL4 cooperates with TLR8, in primary human monocytes, which correspond to cells that migrate into inflammatory sites and are highly relevant for disease pathogenesis. We find that CXCL4 modulates the TLR8 response by synergistically activating TBK1 and IRF5 signaling, and by remodeling the epigenome to regulate transcriptional responses to TLR8 signaling. This results in increased inflammatory gene and inflammasome activation. These findings are relevant for determining the protective versus pathogenic outcomes in conditions, such as viral infections and the autoimmune diseases SSc and systemic lupus erythematosus, where nucleic acid sensing by endosomal TLRs is important.

## Results

**Synergistic activation of inflammatory genes by CXCL4 and TLR8.** To gain a comprehensive understanding of gene regulation by CXCL4 and TLR8, we treated primary human blood monocytes with CXCL4 and the TLR8 ssRNA ligand ORN8L[26,48] for 6 h and performed transcriptomic analysis using RNAseq (Supplementary Fig. 1a). Significantly differentially expressed genes (6518 DEGs; FDR < 0.05, fold change >2) clustered into 5 groups based on pattern of expression (Fig. 1a). Gene groups I and IV were synergistically induced by CXCL4 and TLR8 signaling and included multiple inflammatory genes such as *TNF*, *IL6*, *IL1B*, *IL12B*, *CSF2*, and *IL23A*. Gene-ontology (GO) analysis[49] of genes upregulated by (CXCL4 + TLR8) signaling showed enrichment of innate immune, antiviral and inflammatory pathways (Fig. 1b), and Ingenuity Pathway Analysis (IPA) showed highly significant enrichment of inflammatory pathways in gene groups I and IV (Supplementary Fig. 1b–d). Synergistic induction of *TNF*, *IL6*, and *IL12B* mRNA and primary transcripts (indicative of increased transcription) was confirmed over a time course of CXCL4 and TLR8 stimulation by qPCR (Fig. 1c and

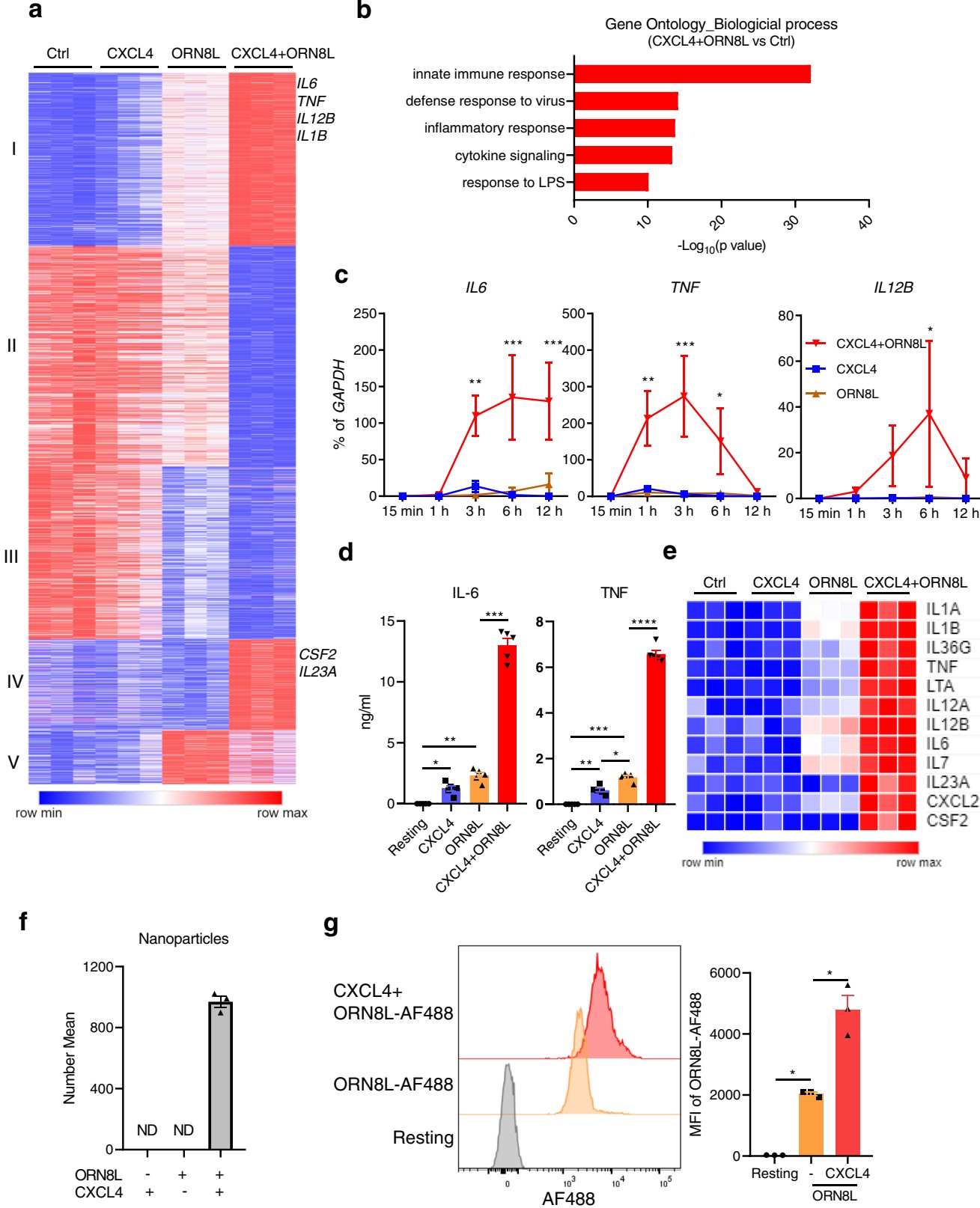

Supplementary Fig. 4a), and ELISA of culture supernatants showed synergistic induction of TNF and IL6 proteins (Fig. 1d). Synergistic gene induction by CXCL4 and cell surface TLR2 and TLR4 was not observed (Supplementary Fig. 2a), which is in accord with the proposed model of CXCL4 interaction with endosomal TLRs[3,6,34,42,43]. Synergistic induction of representative

inflammatory genes by CXCL4 and TLR8 signaling is depicted in a heat map (Fig. 1e), and additional bioinformatic analysis and heat maps of the CXCL4- and TLR8-induced gene response are presented in Supplementary Fig. 2b–e and Supplementary Fig. 3.

One interesting aspect of gene regulation was that at this late 6 h time point CXCL4 attenuated induction of a subset of genes

**Fig. 1 Synergistic activation of inflammatory genes by CXCL4 and TLR8.** Human blood monocytes were stimulated with CXCL4 (10 µg/ml) and/or TLR8 ligand ORN8L (20 µg/ml). **a** RNAseq data analysis. K-means clustering (K = 5) of differentially expressed genes induced more than twofold with FDR < 0.05 by CXCL4, ORN8L, and CXCL4 + ORN8L at 6 h (n = 3 independent blood donors). **b** Gene Ontogeny (GO) analysis of genes upregulated by CXCL4 + ORN8L costimulation versus resting cells. **c** mRNA of *IL6*, *IL12B*, and *TNF* was measured by quantitative PCR (qPCR) and normalized relative to *GAPDH* mRNA. Cumulative data from 4 experiments. **d** IL6 and TNF protein levels in the supernatant of cultured cells were measured by ELISA. Cumulative data from 5 experiments. **e** Heat map depicting expression of representative cytokine and chemokines from (**a**) presented relative to maximum expression. **f** Nanoparticle formation measured using dynamic light scattering. Number mean = the average size of nanoparticles. ND = not detected. The number mean is from one sample measured in triplicate and is representative data out of 3 experiments. **g** Flow cytometric analysis of the internalization of ORN8L-AF488 after 30 min incubation in the absence or presence of CXCL4 in human monocytes. Left panel, representative FACS plot; right panel, cumulative data from 3 experiments. Data was depicted as mean ± SEM; ****$p \leq 0.0001$; ***$p \leq 0.001$; **$p \leq 0.01$; *$p \leq 0.05$ by two-way ANOVA (**c**) and one-way ANOVA (**d**, **g**). Source data are provided as a Source data file.

by TLR8 (Fig. 1a, gene group V). Bioinformatic analysis revealed that the most significantly suppressed pathways were related to interferon signaling and antiviral responses (Supplementary Fig. 1d). A heat map (Supplementary Fig. 3d) depicts attenuated expression of a subset of interferon-stimulated genes (ISGs) although expression of other ISGs was mostly maintained (data not shown). MHC class II genes were also strongly downregulated (Supplementary Fig. 3g). Overall, these results show that CXCL4 and TLR8 cooperate to drive expression of a large number of inflammatory genes, and suggest that costimulation with CXCL4 can modify TLR8 responses to shift the balance to increasing a canonical inflammatory response typically mediated by NF-κB signaling while partially attenuating the IFN response at the late 6 h time point.

It is possible that CXCL4 augments TLR8 responses by increasing TLR8 expression. However, TLR8 mRNA was not increased by CXCL4 and only modestly increased by (CXCL4 + ORN8L) costimulation, and TLR8 protein amounts were only minimally changed (Supplementary Fig. 4b, c). We therefore investigated whether CXCL4 increased ORN8L internalization, similar to a previous study showing that CXCL4 binding of NAs resulted in liquid crystalline complexes and increased internalization in pDCs[6]. Incubation of CXCL4 with ORN8L resulted in nanoparticle formation (Fig. 1f and Supplementary Fig. 4d) and increased ORN8L internalization in primary human monocytes, which increased over time (Fig. 1g and Supplementary Fig. 4e). These data suggest that a chaperone function of CXCL4 that promotes ORN8L internalization contributes to increased TLR8 responses. To address the possibility that low amounts of LPS present in CXCL4 preparations (1–2 pg/ml, Supplementary Fig. 5a) contribute to costimulation of TLR8 responses, we tested the ability of such low subthreshold amounts of LPS (2 pg/ml) and also of a stimulatory amount of LPS (5 ng/ml) to potentiate TLR8 responses. In contrast to CXCL4, LPS at either concentration did not potentiate TLR8-mediated inflammatory gene expression (Supplementary Fig. 5b, c). We conclude that the CXCL4-mediated costimulatory effect is not secondary to LPS contamination and that LPS does not costimulate TLR8 responses in our system.

**Additive and independent activation of NF-κB and MAPK signaling by CXCL4 and TLR8.** We reasoned that the approximately 2- to 4-fold increase in ORN8L internalization by CXCL4 may be insufficient to fully explain the quantitatively much larger superinduction of gene expression. We hypothesized that in addition to increasing ORN8L internalization, CXCL4 also modulates TLR8 signaling to further amplify gene induction. This may occur by two non-mutually exclusive mechanisms: activation of additional signaling pathways by CXCL4 acting at plasma membrane receptors, and increased TLR8 signaling in endosomal compartments secondary to increased avidity of TLR8 engagement by arrayed NA ligands on nanoparticles. We first tested

whether CXCL4 activates NF-κB and MAPK pathways that are important for inflammatory gene activation. Interestingly, CXCL4 when used alone robustly activated NF-κB and MAPK signaling pathways (Fig. 2a, left panel, and 2b). CXCL4 responses were not mediated by its G protein-coupled receptor CXCR3[50,51], as CXCR3 is not expressed in human monocytes[51] (not shown) and blockade of CXCR3 or inhibition of G protein-mediated signaling had no effect on CXCL4 responses (Supplementary Fig. 6a–c and Supplementary Fig. 7a, b). In accord with previous reports that interaction of CXCL4 with cell surface proteoglycans 'presents' CXCL4 to a low-affinity cell surface receptor(s)[45,46,52], cleavage of cell surface proteoglycans chondroitin sulfate and heparan sulfate significantly attenuated CXCL4-induced NF-κB activation (Fig. 2c). The notion that CXCL4 could signal independently of TLR8 and other endosomal TLRs was supported by the resistance of CXCL4-induced NF-κB activation and gene expression to the inhibitor of endosomal signaling Bafilomycin A1 (BafA1), while TLR8 signaling was exquisitely sensitive (Fig. 2d–f and Supplementary Fig. 8a). Activation of NF-κB signaling and gene expression by (CXCL4 + TLR8) was sensitive to BafA1, highlighting the importance of the TLR8 contribution (Fig. 2d–f and Supplementary Fig. 8a). In addition, CXCL4 responses remained intact after digestion of endogenous nucleic acids in our culture system (not shown) and after inhibition of TLR8 using CU-CPT9a (Supplementary Fig. 8b).

The above described results suggest that in primary human monocytes CXCL4 activates signaling pathways that can cooperate with TLR8 signaling to activate gene expression. Comparison of signaling by CXCL4, TLR8, and (CXCL4 + TLR8) revealed additive activation of NF-κB and that CXCL4 is the dominant activator of MAPK signaling (Fig. 2a, b). MAPK signaling was important for synergistic inflammatory gene induction, as inhibition of MAPKs strongly suppressed induction of *TNF* and *IL1B* (Fig. 2g and Supplementary Fig. 8c). Together, these data support the notion that integration of CXCL4 and TLR8 signaling contributes to synergistic gene induction. In this model both CXCL4 and TLR8 activate NF-κB, CXCL4 predominantly drives MAPK signaling, and TLR8 could potentially provide a distinct signal important for inflammatory gene induction, as described below.

**Synergistic activation of TBK1 by CXCL4 and TLR8 drives inflammatory gene expression.** The related kinases TBK1 and IKKε are activated by multiple innate immune receptors, including TLR8, and are most closely linked with activation of IRF3, its downstream target gene *IFNB1*, and subsequent induction of ISGs[24,53,54]. Strikingly, treatment of monocytes with the combination of CXCL4 and the TLR8 ligand ORN8L resulted in a massive increase in TBK1 phosphorylation relative to stimulation with either factor alone (Fig. 3a); increased IKKε phosphorylation was also observed after (CXCL4 + TLR8) stimulation (Supplementary Fig. 9a). As expected (CXCL4 + TLR8) costimulation

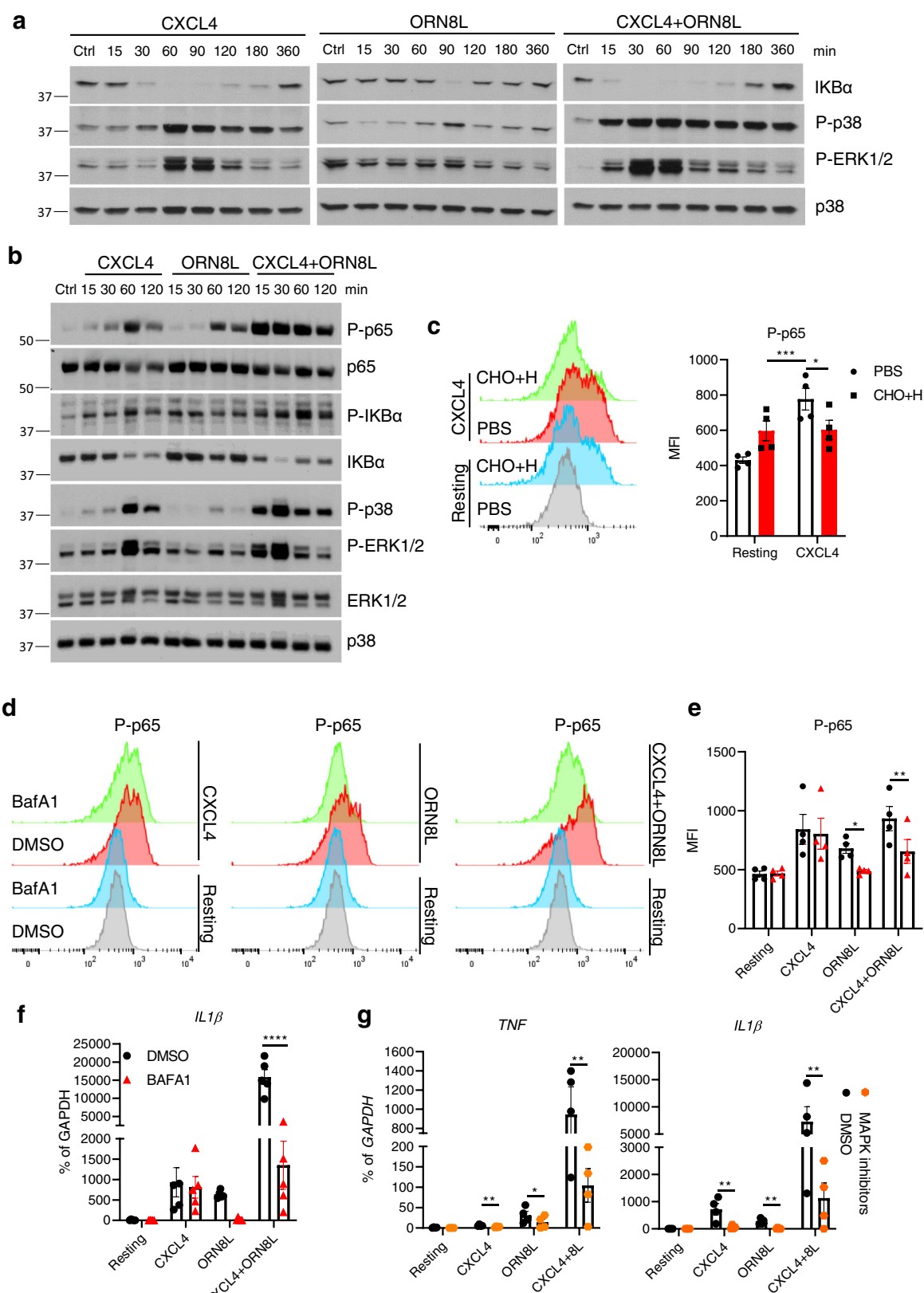

activated IRF3 (Fig. 3b and Supplementary Fig. 9b); synergistic activation of p-IRF3 was not observed but instead (CXCL4 + CXCL8) also induced faster migrating phospho-bands (Supplementary Fig. 9b) that may correspond to cleavage products and IRF3 inactivation as previously reported[55]. (CXCL4 + TLR8) transiently induced *IFNB1* (encoding IFN-β) expression with

peak expression at 1 h and a return to near baseline at the 6 h time point (Fig. 3c) when the above described RNAseq experiments were performed. Induction of the IRF3-*IFNB1* axis provides a functional readout of increased TBK1/IKKε activation; in line with the established predominant role for TBK1/IKKε signaling in IFN responses[24,53,54], the TBK1/IKKε inhibitor

**Fig. 2 Activation of NF-κB and MAPK signaling by CXCL4 and TLR8 in human monocytes. a, b** Immunoblot of whole-cell lysates from monocytes stimulated with CXCL4 (10 μg/ml) and/or TLR8 ligand ORN8L (20 μg/ml) for indicated time course. Data are representative of 4 (**a**, the samples derive from the same experiment and the gels/blots were processed in parallel) or 2 (**b**) experiments. Molecular mass markers are displayed to the left of panels. **c** Flow cytometric analysis of NF-κB p65 phosphorylation in cells pretreated with 1 mg/ml chondroitinase ABC (CHO) and 1 mg/ml heparinase III (H) for 1 h, and then stimulated with CXCL4 for 1 h. Left panel depicts representative histograms and right panel depicts data from 4 experiments. **d, e** Flow cytometric analysis of NF-κB p65 phosphorylation in monocytes pretreated with 1 μM Bafilomycin A (BafA1) for 30 min, then stimulated with CXCL4 and/ or ORN8L for 1 h. **d** Representative histograms and **e** cumulative data from 4 experiments. **f, g** qPCR analysis of mRNA amounts normalized relative to *GAPDH* mRNA in cells stimulated with CXCL4 and/or ORN8L after treatment with BafA1 (**f**) or the combined MAPK inhibitors SB 202190 (p38), JNK inhibitor II and U0126 (MEK1/2) used at 10 μM (**g**). Data in (**g**) are related to and uses some of the same samples as Supplementary Fig. 5c. Data are from 5 (**f**) or 4 (**g**) independent experiments. Data are depicted as mean ± SEM (**c, d–g**). ****$p \le 0.0001$; **$p \le 0.01$; *$p \le 0.05$ by two-way ANOVA. Source data are provided as a Source data file.

MRT67307 suppressed induction of *IFNB1* and the ISG *CXCL10* (Fig. 3d). Surprisingly, MRT67307 also nearly completely abolished (CXCL4 + TLR8)-induced expression of canonical inflammatory and NF-κB target genes such as *TNF*, *IL6*, and *IL12B* and attenuated induction of *IL1B* (Fig. 3e). These results were corroborated using two additional distinct TBK1/IKKε inhibitors (Fig. 3f and Supplementary Fig. 9c). As these kinase inhibitors may have off-target effects, we also used siRNA to knock down TBK1/IKKε expression. Although siRNA-mediated knockdown of TBK1 and IKKε was only partial (Supplementary Fig. 9d), combined knockdown of TBK1 and IKKε significantly decreased inflammatory gene expression (Fig. 3g and Supplementary Fig. 9e). Interestingly, TBK1/IKKε inhibition did not diminish NF-κB activation, as determined by p65 phosphorylation, IKBα phosphorylation, and degradation (Fig. 3h), suggesting that TBK1 regulates a distinct pathway(s).

As inhibition of TBK1/IKKε can increase TLR-induced IL-10 expression[56], which suppresses pro-inflammatory gene expression, we tested the regulation and function of IL-10 in our system. TBK1/IKKε inhibition indeed increased *IL10* expression after CXCL4 and TLR8 costimulation in human monocytes (Supplementary Fig. 9f). Although, as a positive control, blockade of IL-10 signaling using anti-IL-10 and anti-IL10R neutralizing antibodies increased (CXCL4 + TLR8)-induced *TNF*, *IL6*, and *IL1β* expression as expected, TBK1/IKKε inhibition still effectively suppressed inflammatory cytokine gene induction when IL-10 signaling was blocked (Supplementary Fig. 9g). Collectively, the results demonstrate synergistic activation of TBK1/IKKε by (CXCL4 + TLR8), and link these kinases with activation of inflammatory genes.

**Distinct regulation of gene promoters and enhancers by CXCL4 and TLR8.** Effective activation of inflammatory genes by the signaling pathways described above requires remodeling of chromatin and activation of regulatory elements such as promoters and enhancers by signal-induced transcription factors[57]. To gain insight into how (CXCL4 + TLR8)-induced MAPK and TBK1/IKKε signaling is transduced into transcription factor activation and associated chromatin remodeling that enables synergistic gene activation, we performed a genome-wide analysis using ATACseq with footprinting of occupied regulatory elements. Strikingly, CXCL4 alone increased chromatin accessibility (FDR < 0.05, fold change >1.37) at 8391 peaks genome-wide (Fig. 4a, upper left panel); motif analysis under these CXCL4-inducible peaks revealed enrichment of NF-κB and AP-1 motifs (Fig. 4a, lower left panel; ATF3 and BACH2 are members of the extended AP-1/CREB family). This is in line with the signaling results shown in Fig. 2 and suggests that AP-1 transcription factors, which are downstream effectors of MAPK signaling, mediate epigenetic and transcriptional effects induced by CXCL4. TLR8 signaling increased chromatin accessibility at 9707 genomic

regions; in contrast to CXCL4, NF-κB and IRF motifs were most significantly enriched under TLR8-inducible peaks (Fig. 4a, middle panels). This suggested that TLR8 provides an additional IRF-mediated signal, which is consistent with known induction of *IFN* genes, and may complement the CXCL4-activated MAPK-AP-1 axis for gene activation. Costimulation with CXCL4 and TLR8 induced a substantially larger number of 22,517 peaks, with highly significant enrichment of AP-1 and NF-κB motifs, with lesser enrichment of IRF motifs (see next section below for additional discussion) (Fig. 4a, right panels). Analysis of the distribution of ATACseq peaks across the genome and relative to the transcription start site (TSS) revealed inducible peaks in not only promoters but in intronic and intergenic regions (Fig. 4b and Supplementary Fig. 10a). Thus, CXCL4 and TLR8 signaling activate both promoters and enhancers and commonly target gene elements that harbor NF-κB binding sites, with preferential targeting of AP-1 sites by CXCL4 and IRF sites by TLR8.

In accord with the motif analysis, CXCL4 and TLR8 increased chromatin accessibility at overlapping but also distinct regulatory elements (Fig. 4c); 62% of induced ATACseq peaks were common to CXCL4 and TLR8, whereas 21% were specific to CXCL4 and 17% were specific to TLR8. To assess whether differential peak induction could be associated with distinct patterns of gene activation, we used GREAT to associate ATACseq peaks with the nearest genes and performed a KEGG pathway analysis (Fig. 4d). Genes associated with CXCL4- and TLR8-induced ATACseq peaks were enriched in overlapping but also distinct pathways. These results show that CXCL4 and TLR8 can induce distinct open chromatin regions (OCRs) to promote expression of different genes, but also suggest that CXCL4 and TLR8 can cooperatively activate the same genetic elements (overlap area in Venn diagram in Fig. 4c) to augment gene expression. The latter possibility was supported by Formaldehyde-Assisted Isolation of Regulatory Elements (FAIRE) assays showing that CXCL4 and TLR8 signaling can target the *TNF* and *IL6* promoters in an additive (*TNF*) or potentially synergistic (*IL6*) manner (Fig. 4e). Pathway analysis of genes associated with (CXCL4 + TLR8)-inducible peaks showed a complex interaction between the two stimuli that will be described in the next section below.

We extended our ATACseq analysis using TOBIAS (Transcription factor Occupancy prediction By Investigation of ATACseq Signal) to perform footprinting analysis of the open chromatin regions (OCRs)[58]. Relative to the motif enrichment analysis (Fig. 4a) which identifies enriched motifs that have the potential to bind transcription factors (TFs), TOBIAS measures occupancy of precise TF motifs within an ATACseq peak; such footprints show actual TF binding and allow inference of which TF is binding based upon the footprinted sequence. Representative footprints for Rel A (NF-κB p65) and AP-1 family member Fos are shown in Fig. 4f and additional examples are provided in

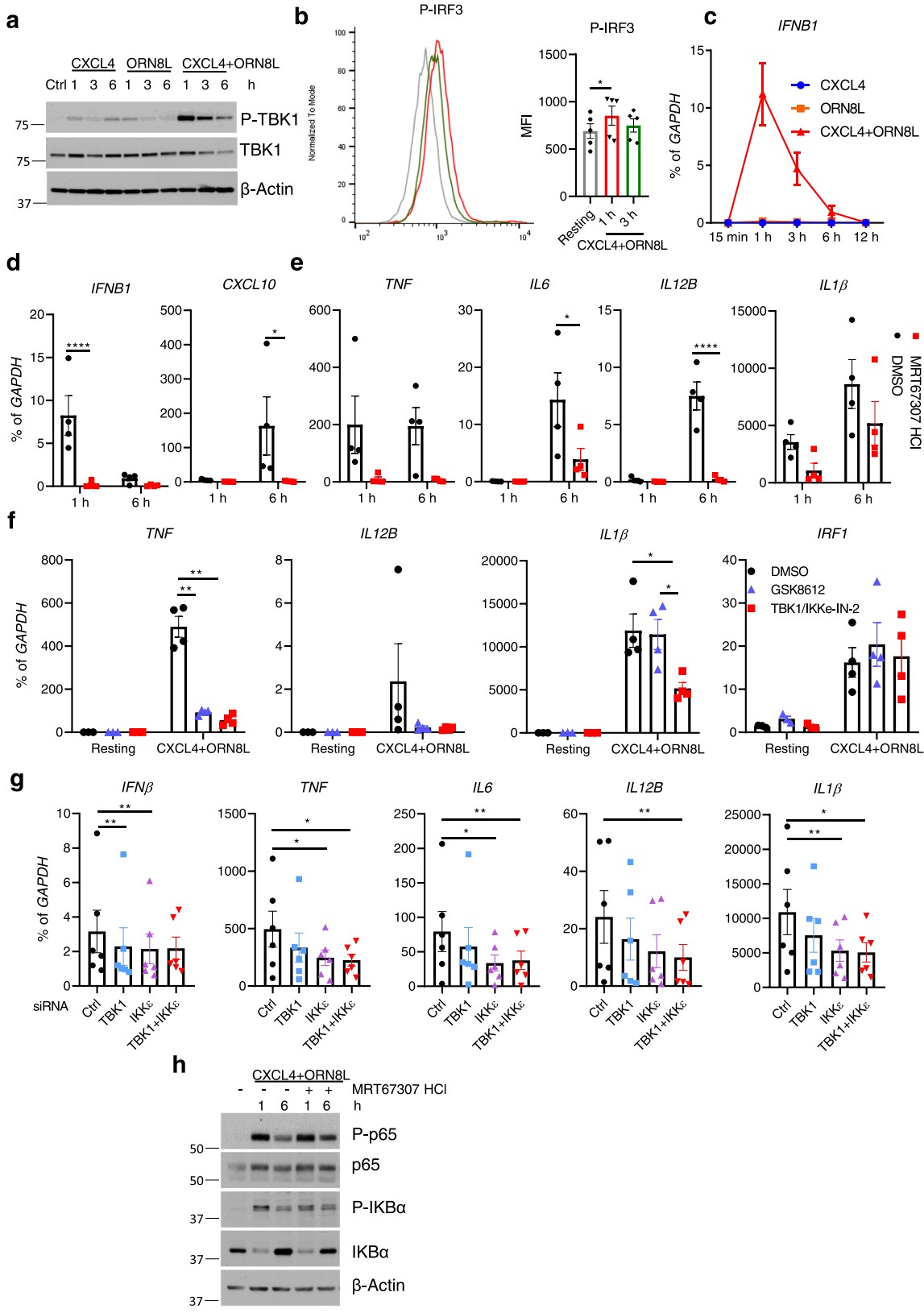

Supplementary Fig. 10b–e. TOBIAS footprinting analysis revealed that CXCL4 strongly and significantly induced occupancy of NF-κB and AP-1 motifs; in addition to canonical Fos and Jun family AP-1 proteins, the footprinted sequences show binding by the BATF, BATF3, and BACH2 members of the extended AP-1 family (Fig. 4g and Supplementary Data 1). In contrast,

TLR8 strongly and significantly induced occupancy of NF-κB and IRF motifs, and also of STAT1:STAT2 and STAT1 motifs (the IRF and STAT binding is consistent with the known TLR-induced type I IFN autocrine loop)[59,60]. These results implicate NF-κB, AP-1, and IRF TFs in mediating chromatin remodeling and gene expression induced by CXCL4 and TLR8, with NF-κB

**Fig. 3 Synergistic activation of TBK1 by CXCL4 and TLR8 drives inflammatory gene expression. a** Immunoblots of phospho-TBK1 and total TBK1 with whole-cell lysates under the indicated conditions. β-actin serves as a loading control. **b** Representative flow cytometry plot (left) and bar graph showing cumulative data (right) of IRF3 phosphorylation after CXCL4 and ORN8L costimulation for 1 h and 3 h ($n = 5$ independent experiments). **c** qPCR analysis of *IFNB1* mRNA normalized relative to *GAPDH* mRNA ($n = 2$). **d**–**g** qPCR analysis of the indicated mRNAs normalized relative to *GAPDH* mRNA. TBK1/IKKε were inhibited using MRT67307 (10 μM) (**d**, **e**), or GSK8612 (50 μM) or TBK1/IKKε-IN-2 (1 μM) (**f**) ($n = 4$ healthy donors), or knocked down using siRNA (**g**). **h** Immunoblot with whole-cell lysates under the indicated conditions. Data are representative of 3 (**a**, **h**), or show cumulative data for 4 (**d**–**f**), or 6 (**g**) independent experiments and depicted as mean ± SEM. ****$p \le 0.0001$; **$p \le 0.01$; *$p \le 0.05$ by Friedman test (**b**, **g**) or two-way ANOVA (**d**, **e**, **f**). Source data are provided as a Source data file.

being activated by both factors, AP-1 predominantly by CXCL4, and IRFs (and STATs) predominantly by TLR8.

Under costimulation conditions, occupancy of AP-1 and NF-κB sites was prominent (Fig. 4g and Supplementary Data 1). Interestingly, binding of IRF sites was diminished under costimulation conditions relative to TLR8 alone, which was consistent with the motif analysis in Fig. 4a. Although initially puzzling, this result is in accord with the attenuation of the IFN response that was observed by RNAseq at this late 6 h time point (Supplementary Fig. 1d). These results suggest that CXCL4 attenuates TLR8-induced IRF binding to OCRs at late time points to modulate the balance between IFN and inflammatory responses.

**Synergistic activation of chromatin by CXCL4 plus TLR8 costimulation.** We next investigated the OCRs that required costimulation by CXCL4 and TLR8 to be induced. In line with the large number of peaks induced by (CXCL4 + TLR8) (Fig. 4a), principal component analysis of all 76,838 ATACseq peaks revealed that OCRs in the CXCL4 + TLR8 condition clearly segregated from OCRs observed under resting or single-stimulation conditions (Fig. 5a).Visualization of the intersections of the CXCL4- and TLR8-inducible ATACseq peaks using an UpSet plot revealed that a large majority of peaks fell into 2 categories: category 2 (C2; 6635 peaks) comprised of OCRs that are induced by both CXCL4 and TLR8 individually and by costimulation, and category 1 (C1; 12,838 peaks) that are induced only when cells were stimulated with both CXCL4 and TLR8 ligand ORN8L (Fig. 5b). Genes associated with C2 peaks showed highly significant enrichment in inflammatory pathways, notably including cytokine–cytokine receptor interaction, antiviral responses, and NF-κB signaling (Fig. 5c). Strikingly, genes associated with C1 peaks were also most significantly associated with the cytokine–cytokine interaction pathway (Fig. 5c). Representative genes associated with C1 peaks in the cytokine–cytokine receptor pathway are displayed in a heat map in Supplementary Fig. 11a and include *IL1B*, *IL6,* and various inflammatory cytokine and chemokine genes. This supports the notion that C1 'synergy peaks' drive expression of a subset of the synergistic gene response. Notably, genes associated with C1 peaks were also significantly enriched in pathways important for cell adhesion and migration, suggesting a novel function for these 'synergy peaks'; representative genes in this category are displayed in a heat map in Supplementary Fig. 11b.

C2 peaks were highly significantly enriched in AP-1 and NF-κB motifs, further supporting cooperative interactions between these transcription factor families (Fig. 5d, e). Strikingly, the most significantly enriched motif in C1 peaks corresponded to the IRF5 binding site (Fig. 5d, e); enrichment of IRF5 motifs in only 5% of targets explains why it was not detected when all inducible peaks were analyzed above. AP-1 and NF-κB motifs were also significantly enriched in C1 peaks (Fig. 5e and Supplementary Data 2). Collectively, the ATACseq results suggest that CXCL4 and TLR8 cooperate to activate regulatory elements in several ways: targeting the same elements to increase their activity,

individually inducing distinct OCRs that can act cooperatively, and synergistically activating an additional set of regulatory elements (C1 peaks) whose activation requires both CXCL4 and TLR8 signals. The results suggest that NF-κB and AP-1 are broadly associated with induction of chromatin accessibility, whereas IRF5 is associated with a subset of synergistically activated regulatory elements. Figure 5f displays representative ATACseq gene tracks at the *IL6*, *TNF*, and *BATF* gene loci that exhibit synergistic induction of open chromatin at regulatory regions.

**CXCL4 and TLR8 costimulation activates a TBK1-IRF5 axis to drive inflammatory gene expression.** In contrast to most IRFs that predominantly activate ISGs and IFN responses, IRF5 can potently activate expression of canonical inflammatory target genes including *IL6*, *TNF*, and *IL1B*[61–63]. Given the enrichment of IRF5 motifs under C1 peaks (Fig. 5d, e), we tested the activation of IRF5 by CXCL4 and TLR8 and its role in synergistic inflammatory gene induction. In line with previous reports[14,15], we found that ORN8L activated IRF5, albeit weakly, as assessed by a shift to a dimeric form detected on nondenaturing gels (Fig. 6a). CXCL4 alone did not activate IRF5, but costimulation by CXCL4 and ORN8L resulted in a massive superactivation of IRF5, with almost a complete shift to the active dimeric form. (CXCL4 + TLR8)-induced IRF5 activation was strongly suppressed by 3 different inhibitors of TBK1/IKKε (Fig. 6a), which is in accord with previous work showing a role for TBK1 in IRF5 activation by TLR8 alone[14,64]. In line with TBK1-mediated activation, IRF5 co-immunoprecipitated with TBK1 after CXCL4 + TLR8 costimulation (Fig. 6b); multiple isoforms of IRF5 were observed as expected[65] (Supplementary Fig. 12a). siRNA-mediated knockdown of IRF5 strongly suppressed (CXCL4 + TLR8)-mediated induction of inflammatory genes including *IL6*, *TNF*, *IL1B*, *IL12B*, and *IRF1*, while minimally affecting expression of the ISG *CXCL10* (Fig. 6c and Supplementary Fig. 12b). These results show that CXCL4 and TLR8 synergistically activate a TBK1/IKKε-IRF5 signaling axis, thereby shifting the profile/balance of TBK1/IKKε function in a more inflammatory direction.

The kinases TAK1 and IKKα/β, best known for activating MAPK and NF-κB signaling, had previously been implicated in IRF5 activation by TLR8 alone[14,64], and therefore we investigated the role of these kinases in (CXCL4 + TLR8) responses. As expected, inhibition of TAK1 or IKK suppressed (CXCL4 + TLR8)-induced activation of NF-κB and inflammatory gene expression (Supplementary Fig. 13a, b); activation of IRF5 was also suppressed (Supplementary Fig. 13c). As expected, these inhibitors had minimal effect on LPS-induced TBK1 activation, but in contrast they suppressed (CXCL4 + TLR8)-induced TBK1 activation (Fig. 6d). These results suggest a role for TAK1 and IKKα/β upstream of TBK1 in (CXCL4 + TLR8) signaling.

**CXCL4 and TLR8 costimulation activates the NLRP3 inflammasome.** In most cell types production of mature IL-1β protein and its release into the extracellular space requires 2 signals—a

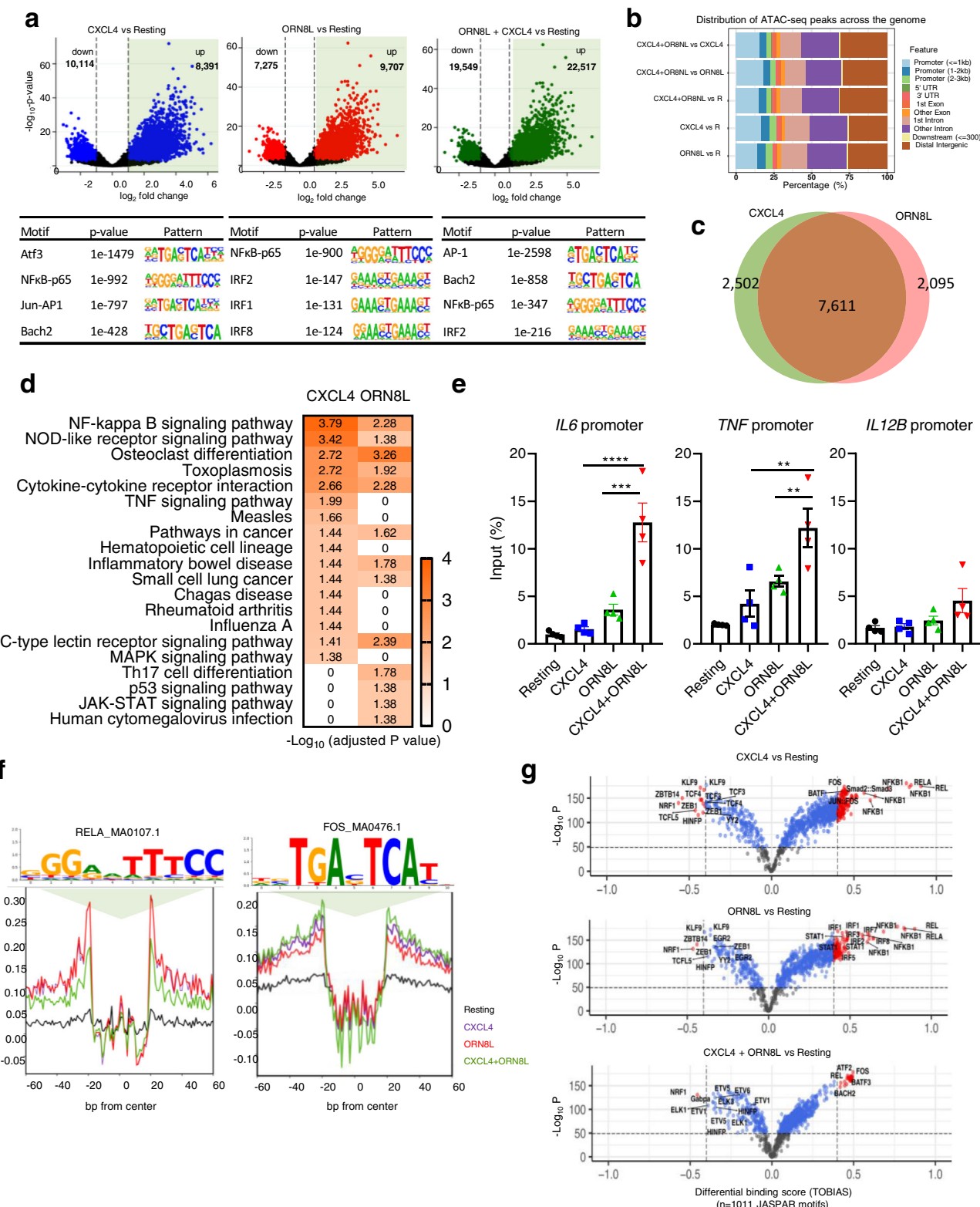

priming signal to activate genes encoding IL-1β and inflammasome components, and a second signal to activate the inflammasome and proteolytic processing of pro-IL-1β into its mature form by caspase 1[66]. As CXCL4 plus TLR8 costimulation strongly induced *IL1B* mRNA (Fig. 1), we tested whether this could result in increased IL-1β protein production. Surprisingly, CXCL4 + TLR8 costimulation, but not CXCL4 or TLR8 alone, induced large amounts of IL-1β protein in monocyte culture supernatants

in a time-dependent manner without the need for a second inflammasome-activating signal (Fig. 7a). In line with these results, the cleaved mature form of IL-1β was detected in cell lysates and culture supernatants of monocytes stimulated with CXCL4 + TLR8 (Fig. 7b). In addition, CXCL4 + TLR8 costimulation induced formation of the cleaved active form of Gasdermin D that is generated by caspase 1-mediated proteolytic cleavage and is required for export of IL-1β from cells (Fig. 7c).

**Fig. 4 ATACseq analysis of induction of open chromatin regions (OCRs) by CXCL4 and TLR8. a** Volcano plots (upper panels) showing differentially induced (right) or suppressed (left) peaks after individual or combined treatment with CXCL4 and ORN8L compared to untreated resting (R) monocytes. DESeq2 Wald test statistic was used to rank the peaks wherein only peaks with Benjamini–Hochberg corrected false discovery rate (FDR) values <0.05 and rank scores above or below 2.5 were considered significant and were chosen for downstream analysis. Lower panels show motif enrichment by HOMER known motif analysis of open chromatin regions. **b** Chart representing the relative distribution of ATACseq peak coordinates across the genome. **c** Venn diagram showing overlapping and distinct ATACseq peaks. **d** KEGG pathway enrichment analysis of genes associated with OCRs. Color scale represents adjusted $p$-value (<0.05), fold change ≥3. **e** FAIRE assays of the *IL6*, *TNF*, and *IL12B* promoter regions 6 h after CXCL4 and/or TLR8 stimulation. $n = 3$ independent experiments. **f** Visualization of RELA_MA0107.1 and FOS_MA0476.1 motif footprints by TOBIAS BINDetect algorithm. **g** Volcano plot depiction of differential binding analysis of $n = 1011$ JASPAR motifs by TOBIAS using BINDetect algorithm. The x-axis represents differential binding score and y-axis −log10 $p$-value. Motifs with significant binding are highlighted in blue and significant motifs with highest differential binding scores in red. ATACseq analysis (**a–d**, **f**, **g**) is based on 3 independent experiments. Data are depicted as mean ± SEM (**e**). ****$p ≤ 0.0001$; ***$p ≤ 0.001$; **$p ≤ 0.01$ by one-way ANOVA (**e**). Source data are provided as a Source data file.

Accordingly, CXCL4 + TLR8 induced formation of the cleaved active form of caspase 1 (Fig. 7d), and inhibition of caspase 1 by AcYVAD decreased the extracellular amounts of IL-1β (Fig. 7e). These results establish that CXCL4 + TLR8 signaling crosstalk activates a caspase 1-Gasdermin D axis leading to the production of mature IL-1β protein in the absence of a distinct inflammasome-activating 'second signal'.

We next investigated whether CXCL4 + TLR8 costimulation activates mature IL-1β production via the canonical NLRP3 inflammasome. CXCL4 alone increased amounts of NLRP3 mRNA, which were further increased when ORN8L was added (Fig. 7f). CXCL4 + TLR8 also synergistically induced NLRP3 protein, which was only detectable by western blotting after costimulation (Fig. 7g). Inhibition of NLRP3 using MCC950 suppressed (CXCL4 + TLR8)-induced cleavage of caspase 1 (Fig. 7d) and decreased extracellular amounts of mature IL-1β (Fig. 7d, h). These results suggest that CXCL4 and TLR8 signaling crosstalk triggers NLRP3 expression and activation in monocytes, which contributes to mature IL-1β protein production. Canonical NLRP3 activation is linked to potassium ($K^+$) efflux from activated cells, which in some systems can be activated by endogenous ATP secreted by activated cells[66–70]. In our system, we found minimal increase of ATP concentrations in the supernatants of (CXCL4 + ORN8L)-stimulated cells (Supplementary Fig. 14a). However, IL-1β release was dependent on $K^+$ efflux, as it was abolished upon addition of $K^+$ to the culture medium (Supplementary Fig. 14b). These data suggest that (CXCL4 + TLR8) stimulation of monocytes is sufficient to induce $K^+$ efflux, although it remains possible that this second signal is triggered by an endogenously produced molecule distinct from ATP. Remarkably, induction of NLRP3 and activation of caspase 1 cleavage were suppressed when TBK1/IKKε were inhibited (Fig. 7i, j). The ability of CXCL4 + TLR8 costimulation, in the absence of an exogenously added trigger of a second signal, to induce inflammasome activation and IL-1β production trended downward upon ex vivo culture and differentiation of monocytes toward macrophages (Supplementary Fig. 14c). Collectively, the results show that synergistic activation of TBK1/IKKε by CXCL4 and TLR8 is coupled to an inflammatory NLRP3-caspase 1-IL-1β pathway in primary human monocytes.

## Discussion

Tight regulation of the magnitude and profile of TLR responses is essential for achieving effective host defense against pathogens while preventing cytokine storm and limiting inflammation-associated tissue damage. In this study, we have identified mechanisms by which CXCL4, a molecule generally considered to be a chemokine, alters the profile of the TLR8 response in human monocytes by dramatically and selectively amplifying TLR8-mediated inflammatory gene transcription and IL-1β production, while attenuating ISG expression at late time points. In addition

to triggering complementary MAPK and IRF pathways, CXCL4 and TLR8 costimulation synergistically activated TBK1/IKKε and repurposed these kinases toward an inflammatory response via coupling with IRF5 and the NLRP3 inflammasome. This integrated CXCL4 and TLR8 signaling was transduced into cooperative and synergistic chromatin remodeling (Supplementary Fig. 15), including de novo formation of unique enhancers with binding of IRF5 motifs, that enabled potent superinduction of inflammatory genes. These findings provide a new paradigm whereby chaperones like CXCL4 modulate TLR responses by cooperative engagement of signaling and epigenomic mechanisms in addition to previously described enhancement of internalization and trafficking of nucleic acids. Cooperative signaling and epigenomic remodeling drive high-level cytokine production that is relevant for pathogenesis of conditions such as RA and potentially COVID-19. Moreover, targeting the TBK1/IKKε-IRF5 axis may be beneficial in inflammatory diseases.

CXCL4 is emerging as a key player in the pathogenesis of SSc, where it has been implicated in fibrosis, and in RA, where CXCL4 is highly expressed in inflamed synovium and has been proposed as one of most informative molecules for differential diagnosis and prediction of disease progression[6,34,37,39]. CXCL4 can promote fibrosis by potentiating TLR9-induced IFN production in pDCs[34] and promoting differentiation of a pro-fibrotic phenotype in monocyte-derived DCs that also become hyper-responsive to eTLR stimulation[42–44]. In DCs, CXCL4 regulates expression of pro-fibrotic genes, which is associated with changes in DNA methylation[44], but signaling pathways that link CXCL4 with fibrogenic genes are not known.

Potentiation of eTLR-induced inflammatory responses by CXCL4 has been attributed to a chaperone function that delivers increased amounts of NA ligands to endolysosomal locations to increase eTLR activation[6], but signaling pathways linking CXCL4 to inflammatory gene induction and synergy mechanisms have not been previously investigated. If CXCL4 worked solely by increasing ligand delivery, it would be predicted to augment standard TLR8 signaling pathways in a TLR8-dependent manner, with a proportionate increase in gene induction. The synergistic gene activation we observed likely additionally requires CXCL4-activated MAPK and NF-κB signaling and chromatin remodeling that occurs independently of TLR8 and endosomal signaling, and synergistic activation of TBK1-IRF5 signaling together with TLR8. Activation of signaling and chromatin remodeling by CXCL4 alone is in accord with the literature suggesting activation of cell surface receptors[36,45,46], which occurs by a proteoglycan-mediated mechanism that was further supported by our results. CXCL4-mediated assembly of nanoparticles containing arrays of CXCL4 and nucleic acid ligands that can extensively crosslink TLR8 with high avidity and amplify signaling also likely contributes to synergistic activation of signaling and opening of chromatin. Furthermore, co-engagement of CXCL4 receptors and

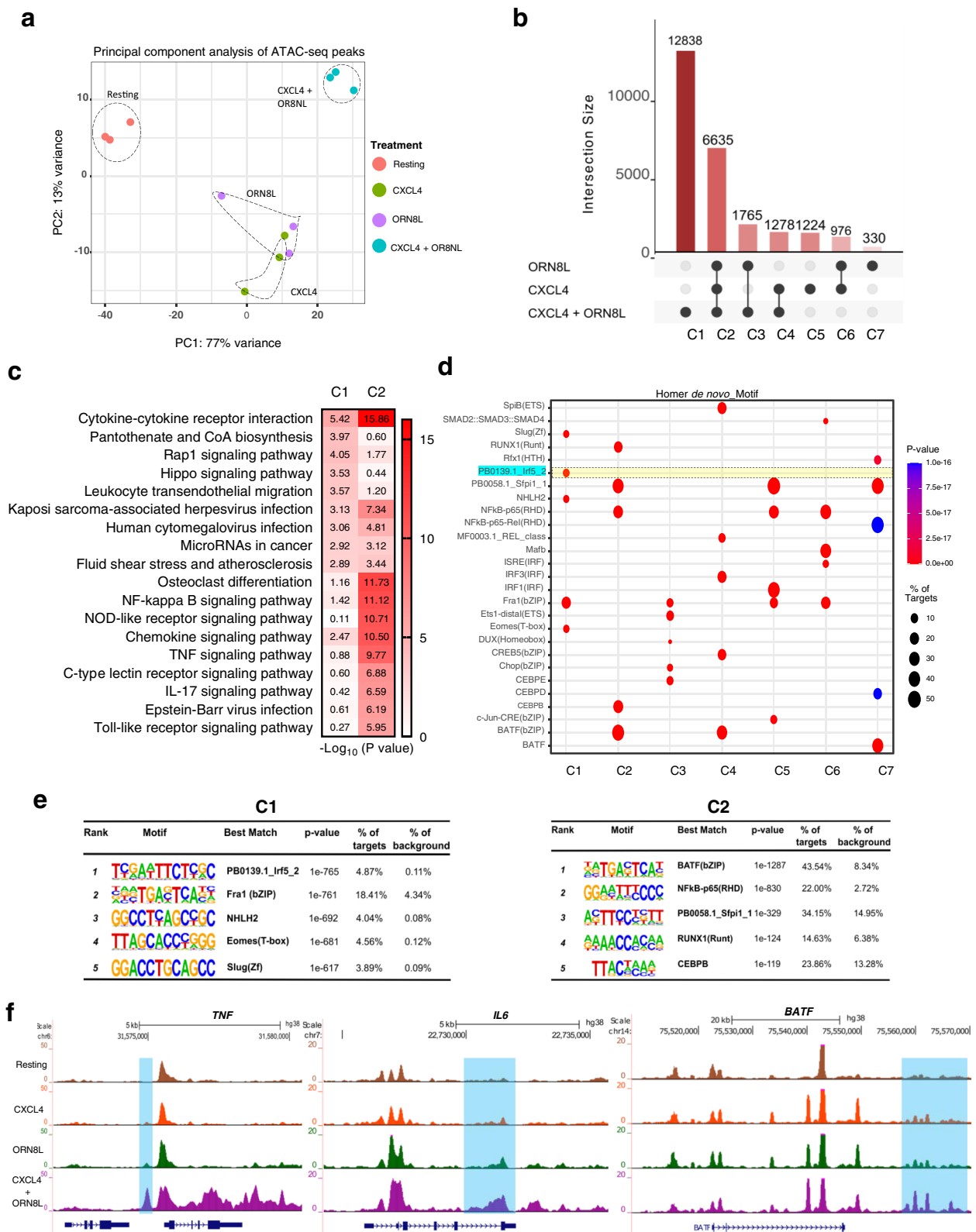

**Fig. 5 CXCL4 and TLR8 costimulation induces unique open chromatin regions. a** Principal component analysis (PCA) of a total of 76,8381 consensus ATAC-peaks for Resting, CXCL4, ORN8L, and (CXCL4 + ORN8L) treatments. **b** UpSet plot identifying distinct and overlapping open chromatin regions. Y-axis depicts the number of peaks and X-axis grouping of peaks by Clusters C1–C7. **c** Functional enrichment analysis by Cistrome-GO of C1 and C2 peaks from (**b**). **d** Dot plot representation of top five motifs identified by HOMER de novo motif analysis of C1–C7 peaks. The dot size represents the percentage of peaks with the corresponding motif, and color represents significance (*p*-value). **e** HOMER known motif analysis showing the top 5 significantly enriched motifs for C1 and C2. **f** UCSC genome browser tracks of normalized ATACseq signal at the *TNF*, *IL6*, and *BATF* loci. ATACseq analysis (**a–f**) is based on 3 independent experiments.

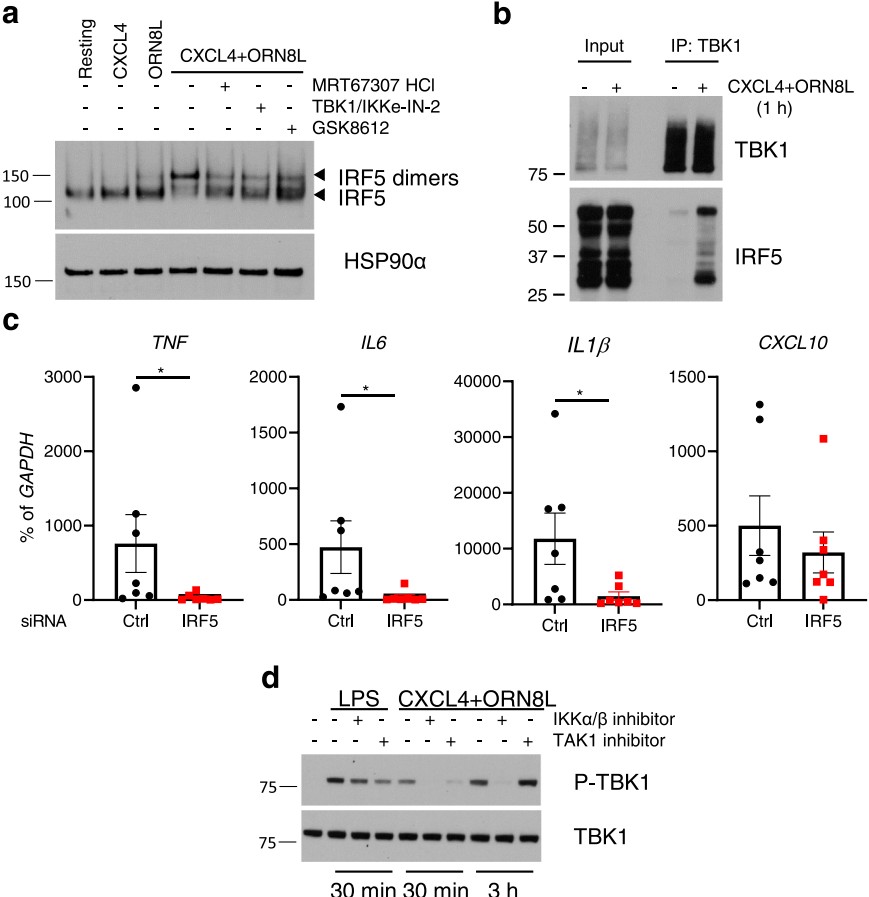

**Fig. 6 CXCL4 and TLR8 costimulation activates TBK1-IRF5 and inflammatory gene expression. a** Immunoblot of IRF5 using whole-cell lysates run on nondenaturing gels. HSP90α serves as loading control. **b** Immunoblot of whole-cell lysates immunoprecipitated with TBK1 antibodies. **c** qPCR analysis of indicated mRNAs normalized relative to *GAPDH* mRNA in monocytes nucleofected with control or IRF5-specific siRNAs ($n = 7$ independent experiments). **d** Immunoblots of phospho-TBK1 and total TBK1 with whole-cell lysates after LPS or (CXCL4 + ORN8L) stimulation for 30 min or 3 h in the presence or absence of IKKα/β inhibitor (BMS-345541, 10 μM) or TAK1 inhibitor (Takinib, 10 μM). Data (**a**, **b**, **d**) are representative of three experiments and depicted as mean ± SEM (**c**). *$p \leq 0.05$ by Wilcoxon signed-rank test, two-tailed (**c**). Source data are provided as a Source data file.

TLR8 by such nanoparticles could also mediate synergistic responses, conceptually similar to amplification of antigen receptor signaling by costimulatory receptors[71]. Important questions to be addressed in future work include identifying specific CXCL4 receptor(s) and understanding how nanoparticle-induced signals are integrated in endolysosomal compartments.

The cooperation of CXCL4 with TLR8 at the signaling and epigenomic levels may be broadly relevant for function of NA-binding chaperones beyond mediating internalization. For example, mechanisms underlying potentiation of eTLR responses by NA-binding autoantibodies, which was described almost 20 years ago[5], have not been fully clarified but likely include signaling in addition to internalization via Fc receptors. Targeting such a costimulation function presents an attractive approach to curtail potentially pathogenic inflammatory responses while leaving aspects of host defense intact.

An important aspect of CXCL4-mediated costimulation is that in addition to activation of complementary signaling and chromatin-mediated pathways, there is a synergistic interaction with TLR8. Robust activation of inflammatory genes required 'coincidence detection' of both CXCL4 and TLR8 ligands (also termed an AND gate), which provides a level of protection from excessive inflammation. In this scenario, recognition of viral RNA by TLR8 prior to substantial tissue damage during infection would contribute mostly to a protective antiviral IFN-mediated response. In contrast, pathogens such as *Staph. aureus* that are

sensed by TLR8[72] and cause extensive tissue damage would induce CXCL4 release from infiltrating platelets or pDCs, thus driving production of inflammatory cytokines and chemokines required for an effective immune response. Similarly, cell death and release of extracellular RNA would induce minimal TLR8 responses unless there is accompanying tissue damage or pDC infiltration and concomitant CXCL4 release. One mechanism underlying synergy is superactivation of TBK1/IKKε, which are coupled to activation of IRF5. IRF5 has the capacity to activate inflammatory genes, can cooperate with NF-κB, and has been closely linked with autoimmune and inflammatory disease pathogenesis[19,61,62].

IRF5 can be phosphorylated by several kinases including TAK1, IKKβ, and Pyk2[73]. In the absence of CXCL4 costimulation, TLR8-induced activation of IRF5 is mediated by TAK1 and IKKβ, with a minor role for TBK1[14,64]; these kinases can phosphorylate IRF5 and/or its upstream adapter TASL. In line with these results, we found a role for TAK1 and IKKβ in IRF5 activation after (CXCL4 + TLR8) costimulation; as expected, inhibition of TAK1 and IKKβ suppressed NF-κB activation and inflammatory gene induction. In contrast to TLR8 signaling alone, TBK1 played a major role in IRF5 activation after (CXCL4 + TLR8) stimulation, possibly related to the function of the CXCL4 receptor(s) or the change in geometry and avidity of TLR8 activation by CXCL4- and RNA-containing nanoparticles. As expected, TBK1 signaling was not required for NF-κB

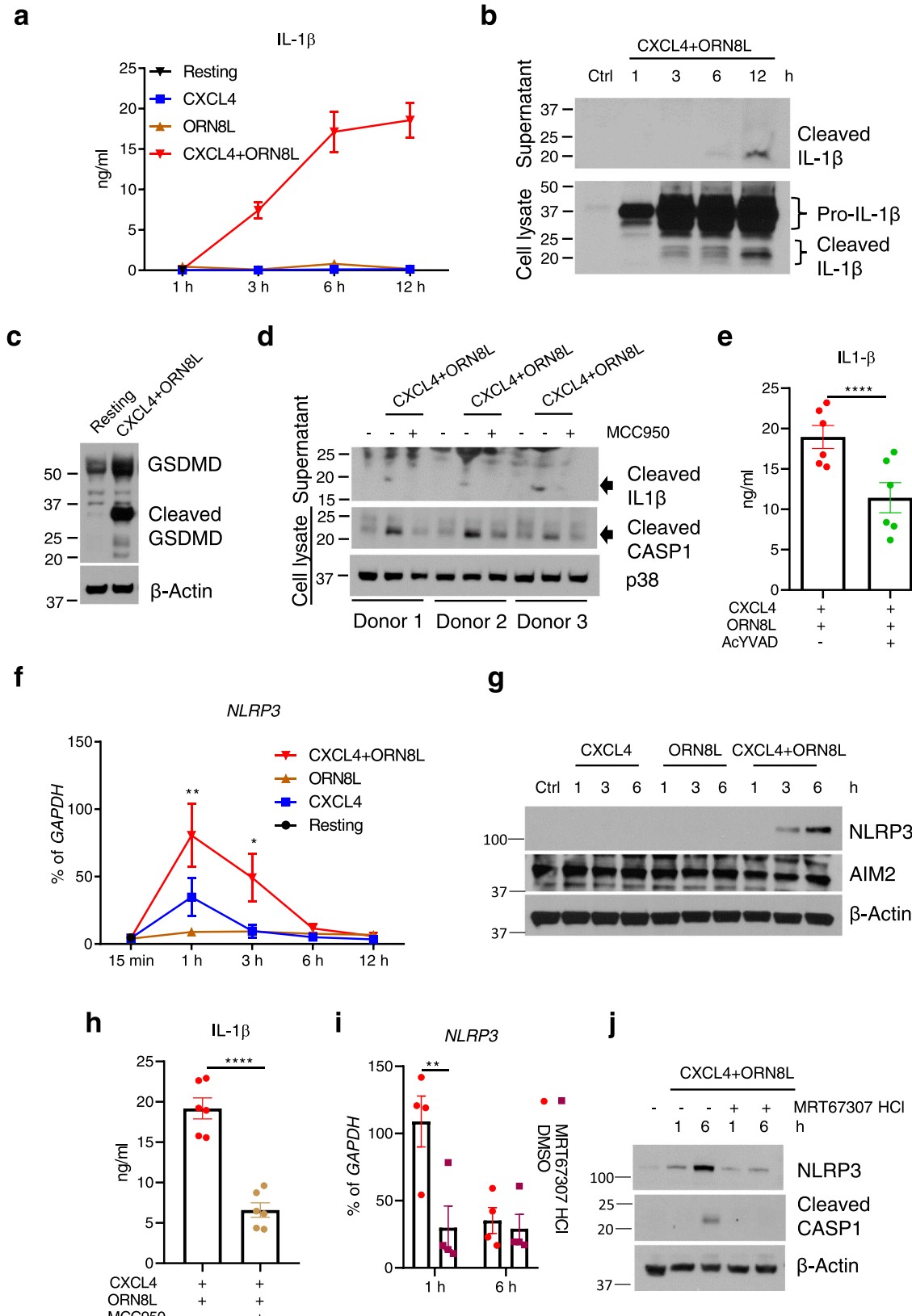

activation but under conditions of costimulation, TBK1 activation was dependent on TAK1 and IKKβ. Thus, costimulation coupled TAK1-IKKβ pathways typically linked with NF-κB activation with TBK1 pathways more typically linked with IFN responses. Although the direct substrates and phosphorylation events mediated by the TAK1, IKKβ, and TBK1 kinases in our system

remain to be determined, the strong coupling of TBK1/IKKε to IRF5 activation by costimulation repurposes these kinases by shifting the balance from an IFN to an inflammatory response at late time points. These results raise the possibility of using TBK1/IKKε inhibitors therapeutically to attenuate eTLR-driven pathogenic inflammatory responses and cytokine storm, as may occur

**Fig. 7 CXCL4 and TLR8 signaling crosstalk activates the NLRP3 inflammasome and IL-1β production. a** ELISA of IL-1β protein in culture supernatants (*n* = 7 independent experiments). **b** Immunoblot of time course of mature IL-1β protein amounts in culture supernatants (top panel) and whole-cell lysates (bottom panel). **c** Immunoblot of GSDMD in whole-cell lysates after 6 h stimulation. **d** Immunoblots of mature Caspase 1 (CASP1) and IL-1β protein in supernatants or whole-cell lysates after 6 h of stimulation with CXCL4 and TLR8 with or without the NLRP3 inhibitor MCC950 (20 μM). Results for 3 different donors are shown. **e** ELISA of IL-1β protein in culture supernatants after inhibition of CASP1 by AcYVAD (10 μg/ml) (*n* = 6 independent experiments). **f** qPCR analysis of NLRP3 mRNA normalized relative to *GAPDH* mRNA in CXCL4 and/or ORN8L stimulated cells in a time-course experiment (*n* = 4 independent experiments). **g** Immunoblots of NLRP3 and AIM2 using whole-cell lysates. **h** ELISA of IL-1β protein in culture supernatants after 6 h of stimulation with CXCL4 and TLR8 with or without the NLRP3 inhibitor MCC950 20 μM (*n* = 6 independent experiments). **i** qPCR analysis of NLRP3 mRNA normalized relative to *GAPDH* mRNA in CXCL4 and ORN8L co-stimulated cells with/without TBK1/IKKε inhibitor MRT67307 HCl pre-treatment (*n* = 4). **j** Immunoblots of NLRP3 and CASP1 using whole-cell lysates. Immunoblot data (**b**, **c**, **d**, **g**, **j**) are representative of three independent experiments and data are depicted as mean ± SEM in the other panels. ****$p \leq 0.0001$; **$p \leq 0.01$; *$p \leq 0.05$ by Paired *t* test, two-tailed (**e** and **h**) and two-way ANOVA (**f** and **i**). Source data are provided as a Source data file.

in RA or COVID-19. This therapeutic strategy would need to be pursued cautiously in primarily TNF-driven diseases or subsets of RA patients, given that TBK1 also restrains RIPK1 and associated cell death[74–76].

CXCL4 and TLR8 signaling are integrated at the level of chromatin remodeling at promoters and enhancers that directly augment gene induction. Reflective of upstream signaling, there are additive and synergistic effects of CXCL4 and TLR8 on chromatin accessibility. In additive interactions, CXCL4 and TLR8 signaling increase opening of chromatin at the same gene locus, or induce distinct OCRs associated with the same gene that can function together to increase gene expression. The contributions of CXCL4 and TLR8 to chromatin remodeling are partially distinct as CXCL4 preferentially targets elements with AP-1 motifs and TLR8 targets elements with IRF motifs, while both signal to open NF-κB motif-containing peaks. Synergy at the level of chromatin remodeling was evidenced by opening of more than 10,000 chromatin regions that required both CXCL4 and TLR8 inputs. These synergy peaks are highly significantly associated with an inflammatory 'cytokine–cytokine receptor' pathway and can further augment expression of canonical inflammatory genes. These peaks are most significantly enriched in the IRF5 binding motif; the ability of IRFs to function as pioneer factors that open inaccessible chromatin suggests a key role for IRF5 in de novo enhancer formation[19]. Interestingly, synergy peaks were also associated with genes in cell adhesion and migration pathways previously suggested to play a role in pro-fibrotic responses[43], and with genes in pathways that regulate cell size and mechanical stress. This suggests that CXCL4 + TLR8 costimulation induces novel macrophage functions that will be interesting to explore in future work.

In addition to synergistic gene induction, CXCL4 and TLR8 signaling cooperated to suppress gene expression (Fig. 1a, gene groups II and III). The downregulated genes were enriched in metabolic pathways including oxidative phosphorylation, fatty acid oxidation and inositol metabolism, and calcium-NFAT signaling (Supplementary Fig. 2b, c). We speculate that the downregulation of these metabolic pathways is consistent with a switch to anaerobic glycolysis that occurs after TLR stimulation, and that downregulation of calcium-NFAT signaling suppresses differentiation of monocytes toward osteoclasts, which is associated with suppression of inflammatory cytokine production. Mechanistically, downregulation of gene expression may be related to suppression of either DNA-binding or expression of transcription factors important for anabolic metabolism and osteoclastogenesis, including SREBPs, TFEB, MITF, MYC, NFATC2, NFATC3, and E2F1 (Supplementary Fig. 3h).

A generally accepted paradigm, established in mouse macrophages and DCs, is that IL-1β production requires a two-step process: a priming step to induce transcription of *IL1B* and genes encoding inflammasome components, followed by inflammasome

activation by a second signal, often provided by ATP, that generates a K+ efflux from cells[77]. Two exceptions to this paradigm have been previously reported, both involving primary human monocytes (but not macrophages). In one model, TLR4 activates the NLRP3 inflammasome via a TRIF-RIPK1 pathway without a classic '2nd signal' and K+ efflux[78]. An alternative model proposes that triggering of cell surface TLRs 2/4/5 induces release from cells of endogenous ATP, which in turn provides the 2nd signal and K+ efflux[79,80]. In our system TLR8 stimulation minimally activated mature IL-1β production, and required CXCL4 to provide signals that induce *IL1B* and *NLRP3* expression, and activate NLRP3-Caspase 1-mediated processing of Gasdermin D and pro-IL-1β. CXCL4 costimulation was sufficient to activate K+ efflux in the absence of addition of an exogenous ligand that activates this "second signal". This activation of K+ efflux may be mediated by an autocrine factor that is distinct from ATP. Activation of Caspase 1 and thus production of massive amounts of mature extracellular IL-1β was dependent TBK1/IKKε, further highlighting the inflammatory role of these kinases under conditions of (CXCL4 + TLR8) costimulation; regulation of NLRP3 by TBK1/IKKε is complex, as these kinases also attenuate second-signal-mediated activation in macrophages[81]. It is possible that high production of IL-1β contributes to the CXCL4-mediated attenuation of TLR8-induced IFN response, as previous work has shown bidirectional negative crosstalk between IL-1β and type I IFNs and the importance of balance between IL-1β and IFNs for *Mycobacterium tuberculosis* pathogenesis[82–85]. Indirect IL-1β-mediated inhibition of the IFN response by CXCL4 would explain why this effect only became apparent at later time points, while the early phase activation of IRF3 and *IFNB1* expression by TLR8 remained intact. Overall, activation of the NLRP3-Caspase 1-IL-1β pathway by CXCL4 costimulation further supports the notion that CXCL4 alters the balance of the functional profile of TLR8 from IFN responses to inflammation, particularly at the later phases of monocyte activation.

In summary, CXCL4 costimulation augments and shifts TLR8 responses in an inflammatory direction by synergistic activation of TBK1 signaling that drives inflammatory outcomes via IRF5-mediated gene induction and inflammasome-mediated IL-1β production, and by genome-wide regulation of chromatin accessibility. These findings suggest that synergistic activation of canonical inflammatory genes after (CXCL4 + TLR8) costimulation can contribute to cytokine storm and inflammatory disease pathogenesis, and can potentially be therapeutically targeted to suppress pathology while preserving aspects of host defense.

## Methods

**Human cells.** Deidentified human buffy coats were purchased from the New York Blood Center following a protocol approved by the Hospital for Special Surgery Institutional Review Board. Peripheral blood mononuclear cells (PBMCs) were

isolated via density gradient centrifugation with Lymphoprep (Accurate Chemical, Carle Place, NY, USA) and monocytes were purified with anti-CD14 magnetic beads from PBMCs immediately after isolation as recommended by the manufacturer (Miltenyi Biotec)[86]. Monocytes were cultured overnight at 37 °C, 5% $CO_2$ in RPMI-1640 medium (Invitrogen) supplemented with 10% heat-inactivated defined FBS (HyClone Fisher), penicillin-streptomycin (Invitrogen), L-glutamine (Invitrogen), and 20 ng/ml human M-CSF. Then, the cells were treated as described in the figure legends.

**Mouse bone marrow-derived dendritic cell (BMDC) culture.** Animal experiments were approved by the Weill Cornell Medicine IACUC Committee. Male C57BL/6J mice (Strain #:000664) at 6–8 weeks old were purchased from the Jackson Laboratories and housed under specific pathogen-free conditions. Bone marrow cells were harvested after euthanasia by $CO_2$ asphyxiation, and cultured in RPMI-1640 medium (Invitrogen) supplemented with 10% heat-inactivated defined FBS (HyClone Fisher), penicillin-streptomycin (Invitrogen), L-glutamine (Invitrogen) and 20 ng/ml mouse GM-CSF and 10 ng/ml mouse IL-4 for 6 days. Then, the cells were treated as described in the figure legends.

**Flow cytometry (FACS) analysis.** Immature floating BMDC were harvested, stained with Fixable Viability Dye eFluor 780 (Thermo Fisher Scientific, 65-0865-18), and then anti-CD11C (BD Biosciences, 749039), anti-Ly6C (Biolegend, 128014), and anti-CD206 (Biolegend, 141716) antibodies for 30 min at 4 degree. For P-IRF3 and P-p65 detection, human monocytes stimulated with CXCL4 and/or ORN8L for the times indicated in the figure legends were fixed with 4% Paraformaldehyde (PFA) in PBS for 15 min at room temperature (RT). After washing with PBS, cells were permeabilized with 0.05% digitonin for 10 min and blocked with 3% BSA for 30 min at RT. Next, cells were stained with anti-P-IRF3-AF488 (1:50, Cell Signaling, 73981S) or anti-p-p65 (1:1600, Cell Signaling, 3033S) antibodies for 2 h. Secondary anti-rabbit-AF594 (1:2000, Thermofisher Scientific, A-11012) antibodies were added to the cells for 30 min after P-p65 antibody staining. After washing, the cells were analyzed using BD FACSymphony A3 Cell Analyzer and Flowjo software.

**RNA sequencing.** After RNA extraction, libraries for sequencing were prepared using the NEBNext Ultra II RNA Library Prep Kit for Illumina following the manufacturer's instructions (Illumina). Quality of all RNA and library preparations was evaluated with BioAnalyser 2100 (Agilent). Sequencing libraries were sequenced by the Epigenomics Core at Weill Cornell using a HiSeq2500, 50-bp single-end reads at a depth of ~20–40 million reads per sample. Read quality was assessed and adapters trimmed using FastQC and cutadapt. Reads were then mapped to the human genome (hg38) and reads in exons were counted against Gencode v27 with STAR Aligner. Differential gene expression analysis was performed in R using edgeR. Genes with low expression levels (<3 counts per million in at least one group) were filtered from all downstream analyses. Benjamini–Hochberg false discovery rate (FDR) procedure was used to correct for multiple testing.

**The ingenuity pathway analysis (IPA).** IPA was used to analyze differently expressed genes. The Ingenuity Canonical Pathways were used to predict activated or suppressed pathways based on the expression pattern of genes regulated by CXCL4 and ORN8L in human primary monocytes. The Upstream Regulator analytic was used to predict upstream regulators whose change in expression or function could explain the observed gene expression changes. The overall activation/inhibition states of canonical pathways and Upstream Regulators are predicted based on a z-score algorithm, for which a negative or positive value represents the predicted inhibition or activation of the pathway and upstream regulator, respectively.

**Formaldehyde-assisted isolation of regulatory elements (FAIRE) assay.** FAIRE assays were performed as previously described[87]. Briefly, cells were cross-linked with 1% formaldehyde for 15 min and quenched with 0.125 M glycine. Then, cells were lysed and sonicated. Ten percent of the samples were used for input and the rest for phenol/chloroform extraction. The input DNA and extracted DNA were used for qPCR. The primer sequences for the qPCR reactions are listed in Supplementary Table 1.

**ATACseq.** ATACseq was performed as described previously[88] with minor modifications. For each condition, a minimum of one million CD14 + human monocytes were collected and centrifuged at 500 × g for 5 min at 4 °C. The cells were washed twice in ice-cold 1x PBS by centrifugation at 500 × g for 5 min at 4 °C. Post-washing the cell pellets were gently resuspended in 50 µl of resuspension buffer (RSB) (10 mM Tris-HCl pH 7.4, 10 mM NaCl, 3 mM $MgCl_2$, 0.1% v/v IGEPAL CA-630, 0.1% v/v Tween-20, 0.01% Digitonin) by gently pipetting up and down 3 times. The lysates were incubated on ice for 3 min, after which 1 ml of RSB buffer containing only 0.1% v/v Tween-20 but not 0.1% v/v IGEPAL CA-630 and 0.01% Digitonin was added. The lysates were centrifuged immediately at 500 × g for 10 min at 4 °C. The supernatants (cytoplasmic contents) were discarded, and the

resulting nuclear pellet was subjected to Tn5 mediated transposition. The nuclear pellet was gently resuspended in a final volume of 50 µl transposase reaction mix (25 µL 2 × TD buffer, 2.5 µL Illumina Tn5 transposase (Illumina Tagment DNA TDE1 Enzyme and Buffer Kits (Illumina, Cat. No: 20034197), 16.5 µl 1X PBS, 0.5 µl 10% Tween-20 (final 0.1% v/v), 0.5 µl 1% Digitonin (final 0.01% v/v), 5 µl nuclease-free $H_2O$), and incubated at 37 °C on a thermomixer set at 1000 rpm for 30 min. The transposed DNA was purified using a QIAGEN MinElute Purification Kit (Qiagen, Cat. No: 28206) and eluted in a 10 µl volume of elution buffer. The transposed DNA was amplified using NEBNext High-Fidelity 2X PCR Master Mix and unique ATAC indexing PCR primers, using the following PCR conditions: 72 °C for 5 min; 98 °C for 30 s; and thermocycling at 98 °C for 10 s, 63 °C for 30 s, and 72 °C for 1 min. The PCR was first performed for a maximum of 5 cycles, after which the optimal number of additional PCR cycles required to avoid variation among samples due to PCR bias was determined by qPCR saturation curve analysis using a 5 µl sample aliquot. Post-amplification, the DNA fragments were size selected by double-sided SPRI bead purification step to remove any PCR primer dimers and large DNA fragments >1000 bp. The amplified DNA libraries containing unique barcode sequences were pooled, and 50-bp paired-end sequencing was performed on an Illumina Hi-Seq 4000 sequencer. A minimum of three biological replicates was included per condition, and at least 35 million paired-end reads were obtained for each sample.

**ATACseq data analysis.** A reproducible ATACseq analysis pipeline TaRGET-II-ATACseq-pipeline (https://github.com/Zhang-lab/TaRGET-II-ATACseq-pipeline) available on a singularity image (ATAC_IAP_v1.1.simg) was used to process ATACseq data. ATACseq read alignments were performed against (GRCh38/hg38) reference human genome. A master consensus peak set comprising a total of 76,838 peaks across four treatment conditions was generated first by requiring that a peak be present in 3 out of 3 replicates per condition, then merging the resulting peak file for each treatment to get the final master consensus peak file. Quantification of peaks to compare the global ATACseq signal changes in the BAM files was done using NCBI/BAMscale program (https://github.com/ncbi/BAMscale). Raw count matrices were obtained using the BAMscale program, and differentially accessible regions compared to the resting condition were identified using the DESeq2 program implemented in the SARTools environment (https://github.com/PF2-pasteur-fr/SARTools). Annotation of the peaks relative to genomic features was done using ChIPseeker[89] and HOMER. Gene Ontology and KEGG pathway analysis was performed using the Genomic Regions Enrichment of Annotation Tool Cistrome-GO and Enrichr using the whole human genome (GRch38/hg38) as the background.

**Transcription factor footprinting analysis by TOBIAS.** Transcription factor Occupancy prediction By Investigation of ATACseq Signal (https://github.com/loosolab/tobias/, TOBIAS, version 0.12.9)[58] was used to find transcription factor footprints in the ATACseq data. Briefly, the merged BAM file of all the three biological replicates per condition was corrected for Tn5 transposase insertion bias using the ATACorrect command. Using the ScoreBigwig command, continuous footprinting scores were calculated on the master consensus peak set of 76,838, resulting in bigWig files with footprint scores. Using the BINDetect command, footprints scores were matched with a list of JASPAR motifs from the JASPAR2020_CORE_vertebrates database followed by calculation of differential binding scores for each motif in the JASPAR database. The differential binding scores and p-values for all the JASPAR motifs highlighting the top 5% and bottom 5% of motifs were shaded in color in volcano plots. PlotAggregate command was used to generate aggregate footprinting plots for any given JASPAR motif sequence for different samples.

**Dynamic light scattering.** CXCL4, ORN8L, or the combination of CXCL4 and ORN8L were diluted to indicated concentration in PBS in a low binding and DNAse and RNAse free tube. 100 µl sample was loaded into a cuvette to measure the nanoparticle polydispersity index (PdI) and diameter of nanoparticles (Number Mean) in a Malvern Zetasizer. Each sample was measured three times at 25 degrees setting on the machine using the automatic measurement duration setting and all samples had to pass the quality control criteria to be recorded, otherwise were noted as not detected.

**ORN8L uptake in human monocytes.** $5 \times 10^5$ human monocytes were plated in 48-well plates and incubated with fluorescently labeled ORN8L (ORN8L-AF488; obtained from Chemgenes Corporation) with/without human CXCL4 for the times indicated in the figure legend. Cells then were harvested, washed with FACS buffer, and analyzed by flow cytometry and the MFI of ORN8L-AF488 was analyzed using Flowjo software.

**Endotoxin detection.** The concentration of endotoxin in CXCL4 stock solutions was measured by Chromogenic LAL Endotoxin Assay Kit (GenScript, cat. No: L00350C) according to the manufacturer's instructions.

**Analysis of mRNA amounts**. Total RNA was extracted with a RNeasy Mini Kit (QIAGEN) and was reverse-transcribed with a RevertAid RT Reverse Transcription Kit (Thermo Fisher Scientific, Catalog number: K1691). Real-time PCR was performed with Fast SYBR Green Master Mix and a 7500 Fast Real-time PCR system (Applied Biosystems). The primer sequences for the qPCR reactions are listed in Supplementary Table 1. CT values of target gene were normalized to *GAPDH* expression and are shown as percentage of GAPDH ($100/2^{\Delta Ct}$).

**Western blotting**. Cells were lysed in 50 μl of cold buffer containing 50 mM Tris-HCl pH 7.4, 150 mM NaCl, 1 mM EDTA, 1% (vol/vol) Triton X-100, 2 mM Na3VO4, 1x phosSTOP EASYPACK, 1 mM Pefabloc, and 1× EDTA-free complete protease inhibitor cocktail (Roche, Basel, Switzerland), and incubated for 10 min on ice. Then, cell debris was pelleted at 13,000 rpm at 4 °C for 10 min. The soluble protein fraction was mixed with 4× Laemmli Sample buffer (BIO-RAD, Cat. #1610747) and 2-mercaptoehanol (BME) (Sigma-Aldrich). Samples for western blotting were subjected to electrophoresis on 4–12% Bis-Tris gels (Invitrogen). To detect IRF5 dimers, nondenaturing Novex WedgeWell 14% Tris-Glycine Gels (Invitrogen, Cat. #XP00140BOX) were used for electrophoresis of protein samples according to manufacturer's instructions. Proteins were transferred to poly-vinylidene difluoride membrane as previously reported[90]. Membranes were blocked in 5% (w/v) Bovine Serum Albumin in TBS (20 mm Tris, 50 mm NaCl, pH 8.0) with 0.1% (v/v) Tween-20 (TBST) at room temperature for at least 1 h with shaking at 60 rpm. Membranes were then incubated with primary antibodies at 4 °C overnight with shaking at 60 rpm. Membranes were washed 3 times in TBST, then probed with anti-mouse or anti-rabbit IgG secondary antibodies conjugated to horseradish peroxidase (GE Healthcare, cat: NA9310V and NA9340V) diluted in TBST at room temperature for one hour with shaking at 60 rpm. Next, membranes were washed 3 times in TBST at room temperature with shaking at 60 rpm. Antibody binding was detected using enhanced chemiluminescent substrates for horseradish peroxidase (HRP) (ECL western blotting reagents (PerkinElmer, cat: NEL105001EA) or SuperSignal West Femto Maximum Sensitivity Substrate (Thermo Fisher Scientific, cat: 34095), according to the manufacturer's instructions, and visualized using premium autoradiography film (Thomas Scientific, cat: E3018). To detect multiple proteins on the same experimental filter while minimizing stripping and reprobing, membranes were cut horizontally based on the molecular mass markers and the molecular size of the target proteins. For membranes that required probing twice or more using different primary antibodies, Restore PLUS western blotting stripping buffer (Thermo Fisher Scientific) was applied on the blots with shaking at 60 rpm for 15 min following first time development. Antibodies used are identified in Supplementary Table 2.

**Immunoprecipitation**. Human monocytes ($20 \times 10^6$) were lysed in 650 μl of IP Lysis/Wash buffer (Pierce Direct IP Kit, Thermo Fisher Scientific, Cat. #26148) and the supernatants of cell lysates after centrifugation were transferred to columns containing the TBK1 antibody linked to AminoLink Plus Coupling Resin to pull down the proteins interacting with TBK1 using the Pierce Direct IP Kit according to the manufacturer's instructions. The immunoprecipitated proteins were denatured with 4× Laemmli Sample buffer and BME and boiled for 5 min, resolved by SDS-PAGE, and immunoblotted with indicated antibodies.

**Cytokine detection by ELISA**. Levels of IL6, TNF, and IL-1β were determined in supernatants of cells using enzyme-linked immunosorbent assay (ELISA) kits (R&D Biosystems) according to the manufacturer's instructions.

**RNA interference**. For RNA interference (RNAi) experiments, primary human monocytes ($6 \times 10^6$ cells) were transfected with 0.1 nmol of siRNA oligonucleotides (listed in Supplementary Table 1) using a Human Monocyte Nucleofactor Kit (Lonza, VVPA-1007) and the AMAXA Nucleofector System (Lonza) program Y001 for human monocyte transfection according to the manufacturer's instructions.

**ATP detection**. ATP concentration in the cell culture medium was determined using ATP Determination Kit (Thermo Fisher Scientific, Catalog number: A22066) following the instructions of the manufacturer.

**IL-10 signaling blockade**. Cells were pretreated with IL-10 and IL10R neutralizing antibodies (10 μg/ml of each antibody) for 1 h and then stimulated with CXCL4 and/or ORN8L in the presence or absence of TBK1/IKKε inhibitor MRT67307 HCl for 3 h. Antibodies used are identified in Supplementary Table 2.

**Statistical analysis**. Graphpad Prism for Windows was used for all statistical analysis. Information about the specific tests used, and number of independent experiments is provided in the figure legends. Two-way ANOVA with Sidak correction for multiple comparisons was used for grouped data; when the data did not pass normality distribution by the Shapiro–Wilk test, the Friedman test with Dunn's correction was used. Otherwise, one-way ANOVA with the Geisser-Greenhouse correction and Tukey's post hoc test for multiple comparisons was performed. For paired data, when the data did not pass the normal distribution by

F test the Wilcoxon signed-rank test was performed, otherwise, paired *t* test was used. Two-tailed tests were used throughout.

**Reporting summary**. Further information on research design is available in the Nature Research Reporting Summary linked to this article.

## Data availability

The RNAseq and ATACseq data generated in this study are available in the Gene Expression Omnibus database under accession code GSE181891. The hyperlink to access the data is https://www.ncbi.nlm.nih.gov/geo/query/acc.cgi?acc=GSE181891. Source data are provided with this paper.

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

## Acknowledgements
We thank the Weill Cornell Medicine Genomics Core Facility for next-generation sequencing, the Weill Cornell Medicine – HSS Flow Cytometry Core Facility for flow cytometry support, and David Oliver (HSS Genomics Center) for advice and discussions. This work was supported by grants NIH R01 AI044938, R01 AR046713, and R01 AR050401 (L.B.I.) and 1R01AI132447 (F.J.B.), and from the Scleroderma Research Foundation (F.J.B.) and from the Scleroderma Foundation (F.J.B.). The David Z. Rosensweig Genomics Center at HSS is supported by The Tow Foundation.

## Author contributions
C.Y. conceptualized, designed, and performed most of the experiments, performed bioinformatic analysis, prepared figures, and wrote the manuscript. M.B. performed the ATACseq experiments, bioinformatic analysis and wrote the manuscript; Y.D. performed the dynamic light scattering experiments; C.B., R.Y., Y.D., M.D.A.K., and G.C. contributed experiments. F.J.B. contributed expertise and intellectual input and edited the manuscript. L.B.I. conceptualized and oversaw the study and edited the manuscript. All authors reviewed and provided input on the manuscript.

## Competing interests
L.B.I. is a nonpaid consultant for Eli Lilly. F.J.B. is a founder of IpiNovyx, a startup biotechnology company. The remaining authors declare no competing interests.
