## [Peer Review File · Nature Communications]

CXCL4 synergizes with TLR8 for TBK1-IRF5 activation, epigenomic remodeling and inflammatory response in human monocytesThis manuscript has been previously reviewed at another journal that is not operating a transparent peer review scheme. This document only contains reviewer comments and rebuttal letters for versions considered at *Nature Communications*.

REVIEWERS' COMMENTS

Reviewer #1 (Remarks to the Author):

.

Reviewer #2 (Remarks to the Author):

The authors made a decent attempt to address the majority of my previous comments for their submission to Nature Immunology. In agreement with the Editors it was concluded that some of the more in depth mechanistic studies that I previously brought up while reviewing the manuscript for its sister journal were not required for publication in Nature Communications. In view of the above, only some minor remaining issues regarding some of my previous comments still need to be addressed:

1. I previously commented that "A clear synergistic induction of TBK1 phosphorylation is shown in fig. 3a, but the downstream effect on IRF3-P (as measured by FACS) seems rather weak. Therefore, the authors should also study IRF3-P via western blot (in fact the same blots as those used for TBK1-P could be used)".

The authors have added IRF3-P western blot data, indicating that they show clear activation (Supplementary Fig. 6b). There is indeed clear activation. However, as far as I can judge, a clear synergy is lacking, at least if one concentrates at the upper band that corresponds to the size of IRF3. Combined treatment seems to result in some clear faster migrating bands that are not visible in the single treatments. However, the size of these bands does not correspond to IRF3. Could these be cleavage products as previously reported in response to TLR3 stimulation (PMID: 21816816)? This would implicate caspase-8 activation and IRF3 inactivation!

2. In reply to my previous comment, the authors show elevated IL-10 levels upon IKKe inhibition but exclude its role in the inhibition of CXCL4/TLR induced proinflammatory gene expression using anti-10 plus anti-IL-10R antibodies. The concentrations used, the source of these antibodies, as well as a reference showing that they do not just bind but also prevent IL-10 and IL-10R activity should be mentioned.

Reviewer #3 (Remarks to the Author):

In this study, Yang and colleagues provide a comprehensive analysis of the epigenomic and transcriptomics downstream effects of costimulation of TLR8 responses by CXCL4, which they are able to link to TBK1-IRF5 signaling. In this revised version of the article, they have addressed the main relevant aspects raised by the reviewers.

From my perspective, the main point of this article is the description of the epigenetic/transcriptomic remodelling downstream to signaling and this is convincingly shown and the quality of the analyses are of quality and provide a highly valuable resource. Of course, there are additional aspects related to the epigenetic control that could have been exploited (particularly given the nature of the pathways involved).

Overall, the reviewer finds the manuscript acceptable for publication.

Response to Reviewers

“CXCL4 synergizes with TLR8 for TBK1-IRF5 activation, epigenomic remodeling and inflammatory response in human monocytes” by Chao Yang, Mahesh Bachu, Yong Du, Caroline Brauner, Ruoxi Yuan, Marie Dominique Ah Kioon, Giancarlo Chesi, Franck J. Barrat, and Lionel B. Ivashkiv.

We are pleased about the positive review of our manuscript and that the Editors are “happy, in principle, to publish a suitably revised version in *Nature Communications*”.

The remaining points of the Reviewers are addressed below, and corresponding changes are highlighted in the manuscript.

Reviewer #2 (Remarks to the Author):

“The authors made a decent attempt to address the majority of my previous comments for their submission to Nature Immunology. In agreement with the Editors it was concluded that some of the more in depth mechanistic studies that I previously brought up while reviewing the manuscript for its sister journal were not required for publication in Nature Communications. In view of the above, only some minor remaining issues regarding some of my previous comments still need to be addressed:”

1. *“I previously commented that “A clear synergistic induction of TBK1 phosphorylation is shown in fig. 3a, but the downstream effect on IRF3-P (as measured by FACS) seems rather weak. Therefore, the authors should also study IRF3-P via western blot (in fact the same blots as those used for TBK1-P could be used)”.*

The authors have added IRF3-P western blot data, indicating that they show clear activation (Supplementary Fig. 6b). There is indeed clear activation. However, as far as I can judge, a clear synergy is lacking, at least if one concentrates at the upper band that corresponds to the size of IRF3. Combined treatment seems to result in some clear faster migrating bands that are not visible in the single treatments. However, the size of these bands does not correspond to IRF3. Could these be cleavage products as previously reported in response to TLR3 stimulation (PMID: 21816816)? This would implicate caspase-8 activation and IRF3 inactivation!”

Response: We thank the Reviewer for this interesting point, which has been added to the text and PMID: 21816816 is now cited as ref. 55 (pg. 8).

2. *“In reply to my previous comment, the authors show elevated IL-10 levels upon IKKe inhibition but exclude its role in the inhibition of CXCL4/TLR induced proinflammatory gene expression using anti-10 plus anti-IL-10R antibodies. The concentrations used, the source of these antibodies, as well as a reference showing that they do not just bind but also prevent IL-10 and IL-10R activity should be mentioned.”*

Response: The concentration and source of the antibodies is now provided in Methods and Supplementary Table 2, as is the information that these have been validated as neutralizing and blocking antibodies. More importantly, in our view, is that In Supplementary Fig. 6g we have provided a positive control that these antibodies work, namely the increase in cytokine mRNA in column 4 versus column 3; this point is made on pg. 9.

Reviewer #3 (Remarks to the Author):

“In this study, Yang and colleagues provide a comprehensive analysis of the epigenomic and transcriptomics downstream effects of costimulation of TLR8 responses by CXCL4, which they are able to link to TBK1-iRF5 signaling. In this revised version of the article, they have addressed the main relevant aspects raised by the reviewers.

From my perspective, the main point of this article is the description of the epigenetic/transcriptomic remodelling downstream to signaling and this is convincingly shown and the quality of the analyses are of quality and provide a highly valuable resource. Of course, there are additional aspects related to the epigenomic control that could have been exploited (particularly given the nature of the pathways involved).

Overall, the reviewer finds the manuscript acceptable for publication.”

Response: We thank the Reviewer for these positive remarks.